# Characterization of the MADS-Box Gene Family in *Akebia trifoliata* and Their Evolutionary Events in Angiosperms

**DOI:** 10.3390/genes13101777

**Published:** 2022-10-01

**Authors:** Shengfu Zhong, Huai Yang, Ju Guan, Jinliang Shen, Tianheng Ren, Zhi Li, Feiquan Tan, Qing Li, Peigao Luo

**Affiliations:** 1Key Laboratory of Plant Genetics and Breeding at Sichuan Agricutural University of Sichuan Province, College of Agronomy, Sichuan Agricultural University, Chengdu 611130, China; 2College of Forestry, Sichuan Agricultural University, Chengdu 611130, China; 3Department of Biology and Chemistry, Chongqing Industry and Trade Polytechnic, Chongqing 408000, China

**Keywords:** *Akebia trifoliata*, MADS-box, whole genome duplication, seed development, angiosperm

## Abstract

As the largest clade of modern plants, flower plants have evolved a wide variety of flowers and fruits. MADS-box genes play key roles in regulating plant morphogenesis, while basal eudicots have an evolutionarily important position of acting as an evolutionary bridge between basal angiosperms and core eudicots. *Akebia trifoliata* is an important member of the basal eudicot group. To study the early evolution of angiosperms, we identified and characterized the MADS-Box gene family on the whole-genome level of *A. trifoliata*. There were 47 MADS-box genes (13 type I and 34 type II genes) in the *A. trifoliata* genome; type I genes had a greater gene length and coefficient of variation and a smaller exon number than type II genes. A total of 27 (57.4%) experienced whole or segmental genome duplication and purifying selection. A transcriptome analysis suggested that three and eight genes were involved in whole fruit and seed development, respectively. The diversification and phylogenetic analysis of 1479 type II MADS-box genes of 22 angiosperm species provided some clues indicating that a γ whole genome triplication event of eudicots possibility experienced a two-step process. These results are valuable for improving *A. trifoliata* fruit traits and theoretically elucidating evolutionary processes of angiosperms, especially eudicots.

## 1. Introduction

Throughout history, plants, especially flowering plants, have evolved ideal molecular mechanisms for morphogenesis, primarily driven by the differential growth of various tissues [1], which is regulated by intertwining network *cis*-element and *trans*-acting factors also called transcription factors [2]. Transcription factors encoded by the MADS-box gene families play fundamental roles in organogenesis control and signal transduction during the tissue growth process and have therefore been extensively studied [3]. In addition, MADS-box genes are also widely used in gene or genome duplication analyses for plant species evolution because they are highly conserved [4]. Thus, further identification of MADS-box gene families from recently published plant genomes is important work.

The MADS acronym is derived from the initial names of the first four proteins with a MADS-box to be reported: MCM1 from *Saccharomyces cerevisiae* [5], AGAMOUS from *Arabidopsis thaliana* [6], DEFICIENS from *Antirrhinum majus* [7] and SRE from *Homo sapiens* [8], which shared the highly conserved 180-bp-long motif called the MADS-box, suggesting that MADS-box genes widely exist in fungi, animals and plants [9]. The conserved MADS domain with approximately 60 amino acid residues at the N-terminus of all MADS-box proteins can bind to the CArG box (CC-A-rich-GC) in the promoter of their target genes [10]. Based on the sequence structure, there are two evolutionary clades of MADS-box genes: type I, including three subgroups (Mα, Mβ and Mγ), and type II, including two subgroups (MIKC* and MIKC^C^). MIKC^C^ can be further divided into the following groups: *SUPPRESSOR OF OVEREXPRESSION OF CONSTANS1/Tomato MADS-box gene 3* (*SOC1/TM3*), *Tomato MADS-box gene 8* (*TM8*), *APETALA3/DEFICIENS* (*AP3/DEF*), *PISTILLATA* (*PI*), *ARABIDOPSIS BSISTER* (*ABS*, *GMM13*), *AGAMOUS* (*AG*)*,*
*AGAMOUS-LIKE12* (*AGL12*), *SEPALLATA* (*SEP*), *SQUAMOSA* (*SQUA*), *ARABIDOPSIS NITRATE REGULATED 1* (*ANR1*), *AGAMOUS-LIKE15 (AGL15*), *SHORT VEGETATIVE PHASE* (*SVP*), *Oryza sativa*
*MADS-box gene 32* (*OsMADS**32)*, *AGAMOUS**-LIKE6* (*AGL6*) and *FLOWERING LOCUS C* (*FLC*) [11]. Although this subfamily has been extensively researched, it is still unclear how many MIKC^C^ groups evolved in plants. Previous studies showed that type II genes experienced a lower rate of birth-and-death evolution than type I genes [4]; according to the ABC model, the identities of floral organs are determined by several subgroups of type II genes, and this was even extended to the ABCDE model [12,13]. Therefore, type II genes have a greater significance than type I genes in the species diversification of angiosperms [14].

Angiosperms consist of approximately 300,000 extant species, more than all other groups of land plants combined [15] and include three major evolutionary branches: basal angiosperms such as *Amborella trichopoda*, monocots such as *O. sativa* and eudicots such as *A. thaliana*. However, eudicots can be further classified into basal eudicots (also called early diverging eudicots or sister groups of core eudicots), such as *Papaver somniferum,* and core eudicots, such as *Vitis vinifera*. The basal eudicots, as the early lineage of eudicots, usually have a parallel evolutionary relative with monocots and are indeed counterparts of monocots. In addition, basal eudicots are an evolutionary bridge between basal angiosperms and core eudicots. This indicates that basal eudicots could have an important role in evolutionary studies.

*A. trifoliata* (Thunb.) Koidz, which is also commonly called augmelon (August melon) in various counties of China because the pericarp rapidly generates a unilateral crevice alongside the ventral suture when its fruit matures [16], belongs to the Lardizabalaceae family of basal eudicots and is a perennial climbing woody liana plant that is widely distributed in East Asia [17]. Practically, it has been widely used as a popular medicinal plant for at least 2000 years [18], and it also has great potential to be exploited as an economic crop, with uses as an edible fruit crop [19], oil crop [20] and ornamental plant [21]; thus, it has recently attracted increasing attention from commercial farmers. Theoretically, *A. trifoliata* is an important member of the basal eudicot group, so it is also of great value in studying the early evolution of angiosperms [22]. Therefore, characterizing the profile of MADS-box gene families and determining the evolutionary relationships among various species will aid in the rapid genetic improvement of economic traits and the realization of organogenesis or tissue formation, such as seed formation.

Four type II MADS-box genes (*AktAP3_1*, *AktAP3_2*, *AktAP3_3* and *AktPI*) were first isolated from *A. trifoliata,* and three *AktAP3* genes could have been putatively produced by two gene duplication events before the origin of the genus *Akebia* [23]. Then, a total of 10 type II MADS-box candidate genes were further identified from this species [22] and all of them were highly conserved and could have been saved by purifying selection during the process of early angiosperm diversification [24]. Although there are some shortages of these studies partly due to the unavailability of genomic data at that time from a systematic and integral standpoint, they afforded much useful information and a reference strategy to further comprehensively understand the characteristics and evolutionary events of MADS-box genes.

In the present study, the available high-quality assembled genome of *A. trifoliata* was employed to accomplish the following objectives: to systematically identify the candidate members of MADS-box gene families, to physically map them on chromosomal positions and to understand their profiles, such as their type, length, number of exons, selection style and putative function. In addition, we would also like to elucidate the evolutionary relatives of MADS-box gene families in whole angiosperms. Ultimately, our findings provide important information on the MADS-box families to further elucidate their functions in *A. trifoliata* as well as in other angiosperms, especially basal eudicots, indirectly contributing to the genetic improvement of commercial traits such as seed yield.

## 2. Materials and Methods

### 2.1. Identification of MADS-Box Genes in A. trifoliata

Genome sequence and annotations of *A. trifoliata* downloaded from the Genome Warehouse of the National Genomics Data Center under the accession number GWHBISH00000000 (https://ngdc.cncb.ac.cn/gwh (accessed on 8 March 2022)) were employed to identify the MADS-box genes. Despite the existence of recent publications on the *A. trifoliata* genome [25], the corresponding genomic data files are still unavailable online. Therefore, a high quality *A. trafoliata* genome assembled and submitted by our group to NCBI was employed the identification and characterization in this study. To identify MADS-box genes, we used the MADS-box typical domain SRF-TF (PF00319) from the Pfam database to search for the predicted genes using the hmmsearch tool in HMMER software (v3.3.2, Sara El-Gebali, UK) with the ‘10e-5 ′ e value parameter [26]. Then, the candidate genes with incomplete domains or gene structure were manually removed. To further confirm the MAD-box feature, the conserved domains and motifs were annotated using the NCBI CDD (v3.19, Aron Marchler-Bauer, USA) and MEME Suite tools (v5.3.2, Timothy L. Bailey, Australia), respectively [27,28]. The MADS-box subfamilies were classified according to sequence similarity and the phylogenetic tree branches of reported MADS-box subfamilies from the reference genome of *A. thaliana* [29]. Another phylogenetic tree was constructed based only on *A. trifoliata* MADS-box genes to further validate the classification results. These two phylogenetic trees were both constructed by using IQtree (v2.2.0.3, Lam-Tung Nguyen, Austria) software with the maximum likelihood method, automatically selecting the optimal substitution model and evaluating branch support values via UFBoot2 tests [30,31,32]. Multiple sequence alignment analysis of MADS-box domains was aligned by ClustalW (v2.0.10, Julie D. Thompson, UK) with default parameters [33]. Previously reported *A. trifoliata* MADS-box genes from the GenBank database and references [22,23] were anchored to our homologues using BLAST (https://blast.ncbi.nlm.nih.gov/Blast.cgi (accessed on 26 March 2022)) with blastn mode.

### 2.2. Chromosomal Distribution of MADS-Box Genes

The *A. trifoliata* genome was previously assembled into 16 chromosomes based on the assembled genome data. The chromosomal positions of the MADS-box genes were assigned using the GFF3 annotation of genome data. The potential cluster of MADS-box genes was identified by sliding window analysis assuming a window size of 250 kb [34]. Then, TBtools (v1.098769, Chengjie Chen, China) was applied to visualize the positions of genes on chromosomes [35].

### 2.3. Synteny and Duplication Analysis of MADS-Box Genes

Both paralogous gene pairs and syntenic blocks of MADS-box genes in the *A. trifoliata* genome were identified using TBtools with default parameters of e-value 1 × 10^−10^ and BlastP hits number 5 for a gene based on the MCScanX tool [36]. A collinear block was defined as at least 5 paralogous gene pairs that were significantly aligned (1 × 10^−5^ as default). Then, the duplication types containing singleton, dispersed, proximal, tandem and WGD/segmental of the MADS-box genes were determined according to the MCScanX synteny analysis results. The nonsynonymous substitution rate (Ka) and synonymous substitution rate (Ks) of MADS-box duplicated gene pairs were calculated using TBtools with the NG method.

### 2.4. Construction of A Phylogenetic Tree of Type II MADS-Box Genes from 22 Species

A total of 22 species genomes (*A. trichopoda*, *N. colorata*, *O. sativa*, *Sorghum bicolor*, *Apostasia shenzhenica*, *V. vinifera*, *A. majus*, *Helianthus annuus*, *A. thaliana*, *Nelumbo nucifera*, *P. somniferum*, *Macleaya cordata*, *A. coerulea*, *Beta vulgaris*, *B. distachyon*, *Populus trichocarpa*, *Theobroma cacao*, *P. patens*, *Selaginella moellendorffii*, *Spirodela polyrhiza*, *Zostera marina* and *A. trifoliata*) were employed to identify MADS-box genes, in which *A. thaliana* [37], *O. sativa* [38] and *A. trichopoda* [39]) were applied as references to identify the subfamilies of the MADS-box genes. The same identified methods as for *A. trifoliata* were used to detect the MADS-box genes for 22 plant species. After classifying the MADS-box subfamilies, the phylogenetic tree of every type II subfamily was constructed and modified using IQtree and FigTree v1.4.4 (https://github.com/4ambaut/figtree (accessed on 21 June 2022)) [30]. The parameter settings were the same as those listed above for the phylogenetic tree of *A. trifoliata* and *A. thaliana* to detect potential duplication events. Gene trees were rooted with genes from mosses or basal angiosperms and cladogram transformation of the branches was applied.

### 2.5. Expression Analysis of A. trifoliata MADS-Box Genes

The 271.49 Gb Illumina transcriptome data of *A. trifoliata* consisted of 36 samples of three tissues (peel, flesh and seed) at four different stages (young, enlargement, colouring and maturity) with three biological replicates, respectively. Details of the data are present in NCBI BioProject PRJNA671772 (accession IDs: SAMN16551931-33, young stage of rind, flesh and seed; SAMN16551934-36, enlargement stage; SAMN16551937-39, coloring stage; SAMN16551940-42, mature stage). The sequences of these samples were aligned to the *A. trifoliata* reference genome using Hisat2 (v2.1.0, Mihaela Pertea, USA) software with default parameters [40]. Then, stringtie [41] was used to evaluate the abundance count of each gene in 36 samples. The gene expression levels in each sample were estimated using fragments per kilobase of transcript per million fragments mapped (FPKM) values according to the abundance count data by using R package DESeq2 (v1.36.0, Michael I Love, Germany) [42]. For differential gene expression, 12 groups of 4 stages and 3 tissues with 3 replicates were calculated as the count matrix. The means of each gene expression in three replicates were normalized to log_2_(FPKM + 1) for visual display. After identifying the MADS-box genes from the whole genome, the expression clusters of these genes were plotted using the R package ggplot2 (v3.3) (https://ggplot2.tidyverse.org (accessed on 28 June 2022)) [43].

## 3. Results

### 3.1. The Number and Type of Identified MADS-Box Genes in the A. trifoliata Genome

A total of 47 MADS-box genes were identified from the *A. trifoliata* genome through HMM analysis (Table 1), in which there was a broad range in the gene length (from 360 bp to 80,167 bp), exon number (from one to 19), isoelectric point (from 4.67 to 11.42), amino acid number (from 81 to 545) and molecular weight (from 9.1 to 62.53) of the putative proteins encoded by the MADS-box genes. The phylogenetic tree consisting of both 108 reference MADS-box genes (consisting of both three and 14 subfamilies of type I and type II, respectively) in *A. thaliana* and 47 MADS-box genes identified in *A. trifoliata* was constructed to classify the MADS-box types in *A. trifoliata*. The phylogenetic tree (Figure 1) showed that 47 MADS-box genes of *A. trifoliata* were unevenly classified into all three subfamilies of type I and 13 subfamilies of type II reference MADS-box genes of *A. thaliana* and only FLC subfamilies of type II in *A. thaliana* were not detected in *A. trifoliata* (Figure 1).

The type I group consisted of 13 genes and the type II group consisted of the remaining 34 genes (Table 1). A comparison analysis found that type I genes have a significantly smaller length and fewer exons than type II genes (Appendix A) at the *p* = 0.01 level, while the differences in isoelectric point, number of amino acids and molecular weight of the putative proteins between them were not significant at the corresponding statistical test level. For example, the largest exon number of type I was only 3 for *AktMγ_3,* while that of type II was up to 19 for *AktSOC1/TM3_2*.

In addition, two reference genes (*AT5G55690* and *AT5G58890* in *A. thaliana*) were classified into Mγ subfamilies rather than the previously classified Mβ subfamilies [29], and three *A. trifoliata* MADS-box genes, *EVM0009117*, *EVM0013722* and *EVM0016918* were the paraphyletic group and classified into the same clade. A comparison analysis of the MADS-box domain of the putative proteins encoded by 44 Mβ and Mγ genes (6 from *A. trifoliata* and 38 from *A. thaliana*) also found that *AT5G55690*, *AT5G58890*, *EVM0009117*, *EVM0013722* and *EVM0016918* seemed closer to the Mγ clade than to the Mβ clade (Appendix A). Conservatively, we classified the three genes into the Mβ/Mγ clade (Table 1).

### 3.2. Phylogeny and Conserved Motifs of MADS-Box Genes

Based on the *A. trifoliata* MADS-box gene phylogenetic tree (Figure 2a), 47 identified MADS-box genes in *A. trifoliata* were also classified as described above, except *EVM0004910* (*AktAG/TM8_4*). Motifs and conserved domains of typical MADS-box family members were further annotated to check the MADS domain. The results showed that 46 out of 47 identified MADS-box genes in *A. trifoliata* had the conserved motifs (motif 1) of MADS proteins, except one *Akt Mβ_1*. Though there was no motif annotated with the MEME Suite in *Akt Mβ_1*, a MADS domain could be detected with NCBI CDD (Figure 2b). In addition, all 13 type I and 9 MIKC* genes contained only the conserved MADS domain, while all 25 MIKC^C^ genes had both MADS and K-box domains, except for *AktSOC1/TM3_2*, which had MADS, LPLAT and EFh domains (Figure 2c). Generally, the coding DNA sequence of type I genes with a smaller gene length had good continuity compared with that of type II genes (Figure 2d). Compared with *A. trifoliata* MADS-box genes from a previous report and the GenBank database, only a few genes (10) related to floral organ development were reported (Appendix A), and most of the genes (37) identified in our study have never been deeply studied before.

### 3.3. Chromosomal Position and Duplication Type of MADS-Box Genes

The physical position of all 47 identified MADS-box genes is ranked in Table 1, and 46 MADS-box genes were mapped to almost all 16 *A. trifoliata* chromosomes except chromosome 6 (Figure 3), while only *AktMIKC*_2* was assigned to the unassembled contig. Chromosomes 2, 3 and 4 carried 6, 7 and 9 MADS-box genes, respectively, while chromosomes 1, 10 and 14 carried only one. According to the definition of gene clusters, the genes on the chromosomes were divided into 41 loci, including 37 singletons and 4 gene clusters (Figure 3). Overall, 10 (21.3%) of the 47 MADS-box genes were distributed into 4 clusters. Most MADS-box genes were collectively distributed on chromosomes 2, 3 and 4, accounting for 22 of 47 that could represent paralogous segments resulting from ancestral polyploidization or tandem duplication events. Further synteny analysis (Appendix A) found that 27 MADS-box genes (marked in red in Figure 3) consisting of *Mα* (2), *Mβ* (1), *Mγ* (3), *MIKC** (5), *AG* (2), *AGL6* (2), *ANR1* (2), *AP3*/*DEF* (2), *SEP* (2), *SOC1*/*TM3* (2), *SQUA* (2) and *SVP* (2) could have been putatively produced by whole genome duplication (WGD) or segmental genome duplication, while only three (*AktMα_4*, *AktMα_6* and *AktAP3/DEF_1*, marked in blue in Figure 3) could have been produced by tandem duplication.

### 3.4. Estimation of Ka/Ks Value

The ratio of the nonsynonymous substitution rate (Ka) to the synonymous substitution rate (Ks) was calculated according to the paralogous pairs of 27 WGD or segmental duplication MADS-box genes, and we detected a total of 19 paralogous pairs among them. All 19 Ka/Ks values of these *A. trifoliata* MADS-box duplicated gene pairs were far less than 1 (Appendix A). The largest Ka/Ks value of the paralogous MADS-box gene pair (*AktMIKC*_3* and *AktMIKC*_9*) was only 0.52, and the Ka/Ks value of *Akt AG_3* and *Akt AG_3* was as low as 0.11. In addition, the Ka/Ks value of the paralogous type II MADS-box gene pair was commonly lower than that of type I (Appendix A).

### 3.5. MADS-Box Genes of Evolutionarily Important Species

The genomes of twenty two evolutionarily important species, two mosses, two basal angiosperms, six monocots, five basal eudicots and seven core eudicots (Figure 4a), were employed to identify and classify the MADS-box genes, in which three species, *A. thaliana*, *O. sativa* and *A. trichopoda,* were used as references. A total of 1469 MADS-box genes with 19 subfamilies were identified, and the average number of MADS-box genes was 67, with a range from 24 to 162 (Appendix A). There was a medium coefficient (R = 0.52, *p* = 0.012) between the MADS-box gene number and genomic size among the 22 species. For type I, the number of *Mα* genes was larger than that of *Mβ* and *Mγ* genes in 17 out 22 species, and for type II, there were more MIKC^C^ genes than *MIKC** genes in all 22 species except *Physcomitrella patens* (Appendix A, Figure 4b). Notably, *EVM0004910* (*AktAG/TM8_4*) of *A. trifiliata* was classified into *TM8* (Appendix A) rather than the AG clade (Figure 1). In addition, there was no Mγ in either moss, no *FLC* in all 22 species except for seven core eudicots and no *OsMADS32* clade in 12 eudicots (Appendix A). Both mosses only contained the *GGM13* of *MIKC^C^* and lacked the remaining 14 subfamilies (Appendix A). We also found that the coefficient of variation (CV) of type I genes was larger than that of type II genes (Figure 4b).

### 3.6. Possible Duplication Events on Different Divergent Angiosperm Branches

Plant MADS-box genes, especially type II MADS-box genes, are widely used in gene or genome duplication analyses; in particular, most duplicated MADS-box genes in *A. trifoliata* were derived from a WGD event (Figure 3 and Appendix A). A total of 15 phylogenetic trees of type II MADS-box gene subfamilies in 22 angiosperm species were constructed, and only the phylogenetic tree of *OsMADS32* was not produced because of the lower number of these genes (Appendix A). Although the trace of these gene duplications could be lost in evolutionary processes, we still detected at least one potential ancestral duplication event in the phylogenetic trees of 15 subfamilies except *TM8* and *AGL6* (Appendix A), in which those of MIKC*, *GGM13*, *ANR1*, *SEP* and *AG* were probably duplicated from ancestral angiosperms or seed plant-wide WGDs (ζ or ε) due to the duplications early in basal angiosperms species, those of *AGL12*, *PI*, *SQUA* and *SVP* supported monocot-specific WGD (τ). Interestingly, similarly to the *A. trifoliata*, other basal eudicots also had duplicated genes in most of subfamilies of type II genes including *MIKC**, *AG*, *ANR1*, *AP3/DEF*, *SEP*, *SOC1/TM3*, *SQUA* and *SVP* which highly overlap with core eudicots (Appendix A). These subfamilies were not only duplicated in the core eudicot lineage. Moreover, *DEF/AP3*, *SQUA*, *SOC1/TM3* and *SVP* possibility derived from a duplication event in early eudicot (ψ) and a core eudicot-wide diploid fusion event (ω) as presented in scenario 2 rather than scenario 1 in Figure 4c (Appendix A).

### 3.7. Differential Expression Analysis of MADS-Box Genes of A. trifoliata in Various Fruit Tissues

The expression of many MADS-box genes in the flesh, seeds and peels showed that all type I genes were expressed at very low levels in all samples (Figure 5a,b). In contrast, many type II genes have a higher expression level. Three type II genes (*AktAG_3*, *AktAG_1* and *AktSEP_3*) exhibited high expression at every stage in all tissues, especially flesh (Figure 5c). In addition, many type II genes also exhibited differential expression levels among different tissues. For example, *AktAG_2*, *AktAGL6_2*, *AktMIKC*_8*, *AktMIKC*_9*, *AktMIKC*_5*, *AktAGL15*, *AktGGM13* and *AktAGL6_1* exhibited high expression levels in seeds and low expression levels in both flesh and peel (Figure 5c). Moreover, *AktMIKC*_8* had a differential expression level among different developmental stages of seeds, with an obvious increase with growth progress (Figure 5a). Because *SEEDSTICK* (*STK*) subfamily are a sister group of *AG* genes, it is difficult to differentiate them based solely on a phylogenetic tree. We found that the *AktAG_2* gene was relatively highly expressed in all seed stages, while *AktAG_1* and *AktAG_3* genes were expressed at relatively lower levels in seed. Therefore, *AktAG_2* should be classified as part of *STK* functional genes in seed development.

## 4. Discussion

The name of MADS-box transcription factors indicates that they could exist widely and be highly conserved in plants, animals and fungi, which has been confirmed by various studies [45], and some homology has even been found in the common ancestor to prokaryotes and eukaryotes [46]. In plants, MADS-box transcription factors have evolved important regulatory functions in various biological processes, such as disease resistance [47], signal transduction [48] and morphogenesis, especially flower formation and seed development [49]. MADS-box genes were identified from all plant genomes, but whether their number is related to evolutionary divergence or genomic size was unclear.

Previous studies have shown that the number of MADS-box genes varies slightly among different species and usually ranges in the dozens, although up to 300 have been identified in the extreme case of Wheat [50]. The numbers were 70, 65, 47 and 90 in the genome basal angiosperm *Nymphaea colorata* [51], monocot *Sorghum biocolour* [52], basal eudicto *Aquilegia coerulea* [53] and core eudicot *V. vinifera* [54], respectively, and they were very close to the corresponding numbers (Appendix A). This indicated that the number of MADS-box genes could be slightly related to the evolutionary clade. In the present study, we identified the numbers of these taxa in the genomes of 25 species from five different evolutionary clades: mosses, basal angiosperms, monocots, basal eudicots and core eudicots (Appendix A). Although those of moss were obviously fewer than those of all angiosperms, their variation was slight among the four clades within angiosperms, in which their variation was smaller in monocots than in eudicots. This result further supported the view that the number of MADS-box genes was not usually related to the position on phylogenetic trees of species, and only a dozen genes indicated that they could regulate various biological processes. In addition, previous reports showed 71, 81, 162 and 300 MADS-box genes in 430 Mb *O. sativa* [55], 475 Mb *V. vinifera* [56], 2870 Mb *P. somniferum* [57] and 1700 Mb Wheat [58] genomes, respectively, which suggested that the greater the number of MADS-box genes was, the larger the genomic size was. In this study, we found that the correlation coefficient between the MADS-box gene number and genomic size was 0.52 (Appendix A), indicating a moderate relationship. Therefore, we agree with the view that the number of MADS-box genes could be generally related to genomic size and scarcely related to species divergence.

Here, we identified 47 MADS-box genes from the 619 Mb *A. trifoliata* genome (Table 1), of which 10 were previously reported [22,23] and 37 were newly identified by genome-wide analysis (Appendix A) which were classified into all three subfamilies of type I and 13 subfamilies of type II MADS-box genes when only the *A. thaliana* genome was used as the reference (Figure 1), while there were 14 subfamilies of type II genes when *A. thaliana*, *O. sativa* and *A. trichopoda* were used as the reference (Appendix A). Our comparison analysis found that *EVM0004910* in the *AG* lineage (Figure 1) was reclassified into *TM8* (Appendix A). This could be explained by the fact that there was no *TM8* clade in *A. thaliana* [29]. Therefore, there were a total of seventeen subfamilies, and only two subfamilies, *OsMADS32* and *FLC*, were not identified in the MADS-box genes of the *A. trifoliata* genome (Table 1).

Structurally, type I genes encode only one simple SRF-like MADS domain, containing one to two exons, whereas type II genes encode MEF2-like or MIKC-type MADS domains with multiple additional domains, such as the K-box (keratin-like domain), I-box (intervening domain) and variable C-terminal domain (C-terminal region) [59], and evolutionarily, type II genes are more conserved than type I genes [4]. In type I genes, *AktMγ_3* has three exons, and the other has one or two exons, while the exon number in type II was largely variable (Table 1). We found that the average amino acid number encoded by type I and type II genes was close, while the exon number of type II was significantly greater than that of type I (Appendix A); additionally, the DNA sequence of type I had good continuity compared with that of type II (Figure 2d). Similar results have been reported in grape [54], rice [37] and *B. distachyon* [60].

The number of MADS-box genes was small, but the subfamilies, especially those of type II genes, were very numerous (Appendix A), which indicated that they could have experienced different evolutionary events, such as different genomic duplications and natural selection styles. In addition, despite the small number of MADS-box genes, they were assigned to all 16 chromosomes except chromosome 6 (Figure 3). Both the rich subfamilies and wide chromosomal distribution suggested that whole genomic duplication (WGD) or segmental duplication could be important duplication styles for *A. trifoliata* MADS-box genes. By performing a synteny analysis, we found that 27 (57.4%) out of the 47 total identified *A. trifoliata* MADS-box genes experienced WGD or segmental duplication events. Moreover, all Ka/Ks values were far less than 1 (Appendix A), which clearly indicates purifying selection. Thus, we propose that MADS-box genes in *A. trifoliata* mainly experienced WGD or segmental duplication and strong purifying selection in the long evolutionary period.

Previous studies have suggested that type II genes exhibit higher conservation than type I genes [4]. In the present study, we found that the coefficient of variation (CV) of all three subfamilies, *Mα*, *Mβ* and *Mγ,* of type I genes was greater than 0.8, while that of both *MIKC** and MIKC^C^ of type II genes was less than 0.5 among 22 species (Figure 4a). In addition, although *AT5G55690* and *AT5G58890* were previously classified into *Mβ* (Table 1), the amino acid sequence encoded by them has a high similarity with *Mγ* (Appendix A), which provides a reasonable explanation for why they were classified into the *Mγ* clade in Figure 1 and indicated that there was a rapid evolution from *Mβ* to *Mγ*. Both results confirmed that type II genes were highly conserved. Type II genes also played a key role in investigating the species diversification of flowering plants because the ABC flowering model was commonly regulated by several subfamilies of type II [12,14], even some subfamilies of type II genes involved in specification of ovule and flower development in seed plants were significant expansion in ferns [61], so we focused on the phylogenetic tree analysis for subfamilies of type II (Appendix A).

Only *GGM13* subfamilies of *MIKC^C^* were identified in both moss species (Appendix A), which indicated that these genes could have originated early, generally before basal angiosperm divergence. In contrast, *FLC* subfamilies were only detected in core eudicots (Appendix A) and were putatively core eudicot-specific, which suggested that they could have recently originated, after core eudicots diverged from basal eudicots. Previous studies have suggested a series of WGDs in angiosperms, mainly including ancestral angiosperms or seed plant-wide WGDs (ζ or ε), monocot-specific WGDs (τ) and core eudicot-specific γ WGTs [44,62,63]. In our study, traces of ζ or ε could be detected in the phylogenetic tree of *DEF/AP3*, *SQUA*, *SOC1/TM3* and *SVP*. On the one hand, there were at least two copies of their homologues in many species among 20 angiosperms, especially in *A. trichopoda*, a model basal angiosperm (Appendix A). On the other hand, their topological structures are largely consistent with the early angiosperm species phylogeny (Appendix A). Likewise, traces of monocot-specific τ could also be detected in the phylogenetic tree of the *AGL12*, *PI*, *SQUA* and *SVP* subfamilies (Appendix A). Only the traditional γ-WGT would be challenged by the phylogenetic tree of some subfamilies of type II genes.

The traditional view is that the γ event occurred approximately 117 million years ago (Mya) and at the early stages of core eudicots after the divergence of the Ranunculales and core eudicots [57,63]. In fact, both the composition and occurrence time of the γ event are still in doubt. For example, a shared WGD event on the common ancestor of basal eudicot, which is phylogenetic close to the core eudicot γ-WGT was also detected in *A. trifolita* and many other basal eudicots based on the transcriptome data [61]. However, another genome analyses of *A. coerulea* provided some evidence suggesting that the γ-WGT event may have consisted of two steps [64]. Obviously, precisely positioning the γ-WGT within a narrow time window is difficult mainly due to the following two reasons: one is the absence of knowledge about the WGT, such as whether a single event or two independent events resulted in the triplication of the genome, and the other is the short time space from eudicot divergence to core eudicot divergence. The phylogenetic trees of *DEF/AP3*, *SQUA*, *SOC1/TM3* and *SVP* in our study supported one common eudicot-wide duplication event (ψ) and core eudicot-wide additional diploid fusion event (ω), putatively occurring before and after core eudicot divergence from basal eudicots, respectively (Appendix A), which could be explained well by scenario 2 in Figure 4c (Figure 4c and Appendix A). Furthermore, the genes of the *FLC* and *AGL15* subfamilies were only duplicated in the core eudicot lineage, and two clades of homologues were also more likely to be duplicated from a potential single ω diploidization event rather than an indivisible γ triploidization event (Figure 4c and Appendix A). Parts of type II MADS-box gene subfamilies, such as *AP3*/*DEF*, *SEP*, *SOC1/TM3* and *SQUA,* were considered to be duplicated from core eudicot-wide γ-WGT events in previous reports [14]. Unfortunately, MADS-box genes were not identified completely due to the use of transcriptomic rather than genomic data. For example, only a few MADS-box genes have been previously identified in *A. trifoliata* (Appendix A) [22,23] and in *A. coerulea* (Appendix A) [53]. Importantly, previous phylogenetic analyses of other basal eudicots also afforded some strong evidence of two close duplication events supporting the ψ and ω hypothesis [65]. Comprehensively, the phylogenetic trees of these subfamilies agreed well with scenario 2 rather than scenario 1 of Figure 4c, which suggested that γ could be a multiple event consisting of eudicot-wide ψ and core eudicot-specific ω rather than a single γ event. Although the origin of the γ-WGT event is currently a highly-debated field which cannot be solved simply by using the phylogenetic gene tree of the MADS-box family, our results provide an important clue to solve this problem in the future.

Differential expression analysis using comparative transcriptomics can provide important information about gene functions. We found that many MADS-box genes, especially type I genes, were expressed at extremely low levels in all three tissues (flesh, seed and peel) at all developmental stages, while *AktAG_3*, *AktAG_1* and *AktSEP_3* exhibited high expression in almost all samples (Figure 5), which indicated that only a few MADS-box genes could meet the regulatory requirement of fruit formation and that MADS-box gene expression is usually organ- or tissue-specific. Some previous reports suggested that some genes of both the *AG* and *SEP* subfamilies could be related to fruit development [66,67]; therefore, we concluded that the three genes could regulate whole fruit development. In addition, we also found that eight genes of five subfamilies (MIKC*, *AG*, *AGL6*, *AGL15* and *GGM13*) were only expressed a high level in seeds, and in all subfamilies except *AG15* which existed in all twenty angiosperms (Appendix A). There are various reports stating that MIKC* [68], *AG* [67], *AGL6* [69] and *GGM13* [70] participate in regulating seed development. Therefore, *AktAG_2*, *AktAGL6_2*, *AktMIKC*_8*, *AktMIKC*_9*, *AktMIKC*_5*, *AktAGL15*, *AktGGM13* and *AktAGL6_1* positively regulated seed development in *A. trifoliata.*

## 5. Conclusions

We identified 47 nonredundant MADS-box genes in the *A. trifoliata* genome, and 46 of them were physically assigned to 16 high-quality assembled pseudochromosomes. All 47 genes were classified into 17 subfamilies, and many of them putatively experienced WGD or segmental duplication and purification in the evolutionary process. Several candidate MADS-box genes involved in fruit or seed development were screened. Importantly, traces of major WGD events in whole angiosperms could be detected in many subfamilies of type II genes. Interestingly, the phylogenetic tree of the *DEF/AP3*, *SQUA*, *SOC1/TM3*, *SVP*, *FLC* and *AGL15* subfamilies provides some evidence that γ events are multiple events rather than a single event consisting of a eudicot-wide ψ and core eudicot-specific ω. Comprehensively, both new data resources and insight into traditional γ events would be helpful to practically improve the objective traits of *A. trifoliata* fruits and to completely elucidate the evolutionary process of early-stage eudicots.

## Figures and Tables

**Figure 1 genes-13-01777-f001:**
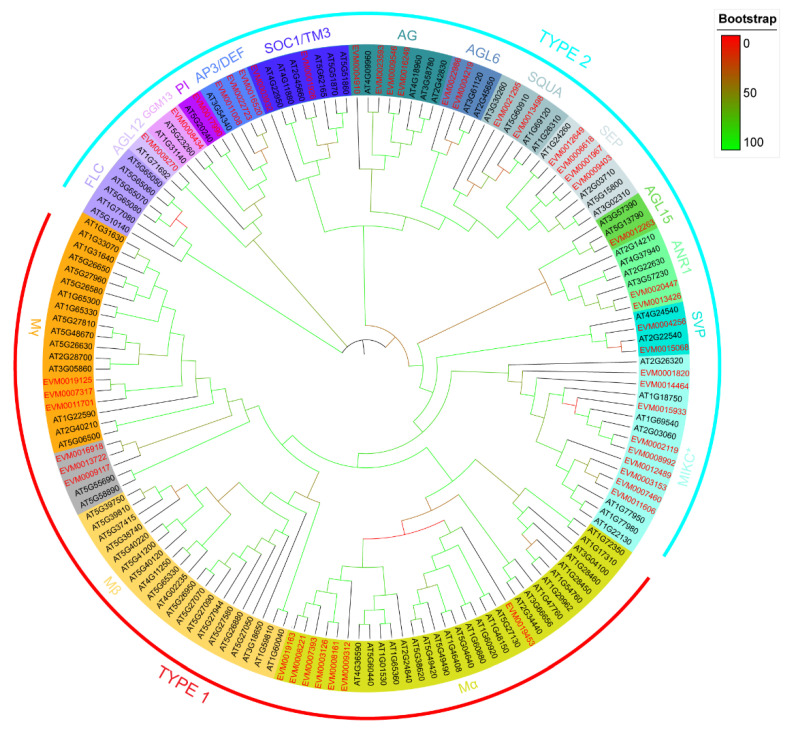
Classification of the *A. trifoliata* MADS-box subfamilies. Specific lineages are indicated by colours and bracketing. The type I lineages contain Mα, Mβ, and Mγ, while the type II lineages contain MIKC* and MIKC^C^ (all other type II subfamilies except MIKC*). The paraphyletic groups of 5 genes, including *EVM0009117*, *EVM0013722*, *EVM0016918*, *AT5G55690* and *AT5G58890,* are coloured grey because *AT5G55690* and *AT5G58890* have previously been classified as Mβ but are placed with Mγ in our analysis.

**Figure 2 genes-13-01777-f002:**
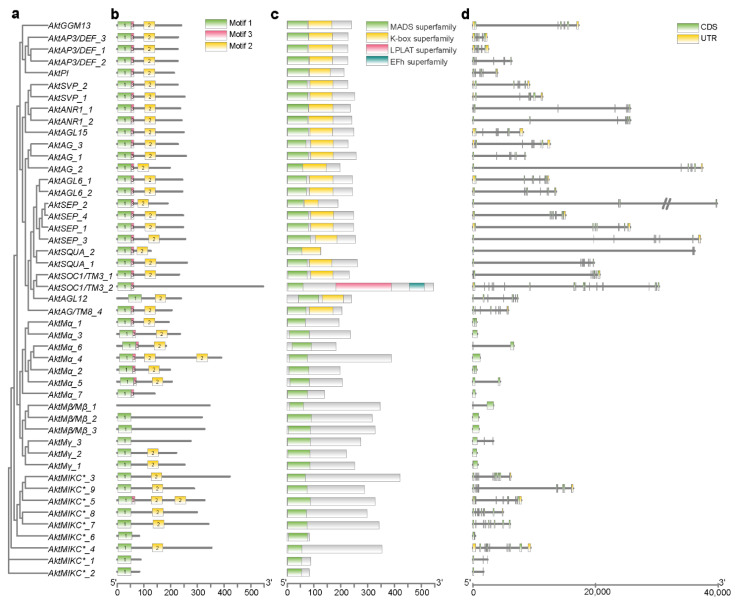
Gene and protein structures of the MADS-box gene families in *A. trifoliata*. (**a**) Phylogenetic tree of *A. trifoliata* MADS-box genes; (**b**) Motifs of MADS-box proteins. (**c**) Conserved domains of MADS-box proteins; (**d**) Exon-intron structures of MADS-box genes.

**Figure 3 genes-13-01777-f003:**
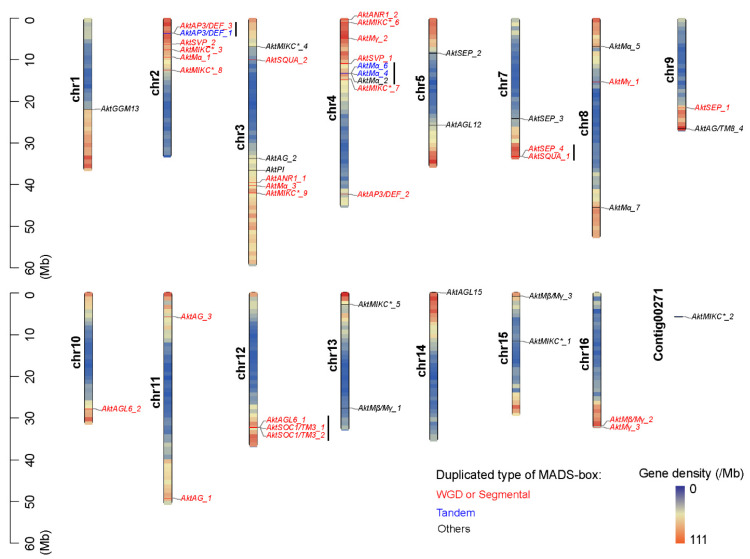
Physical position and duplicated type of MADS-box genes in the *A. trifoliata* genome. WGD or segmental duplicated-type genes are marked in red font, tandem duplicated-type genes are marked in blue font, and other dispersed or proximal duplicated genes are marked in black font. The black line to the right of the gene names represents cluster loci.

**Figure 4 genes-13-01777-f004:**
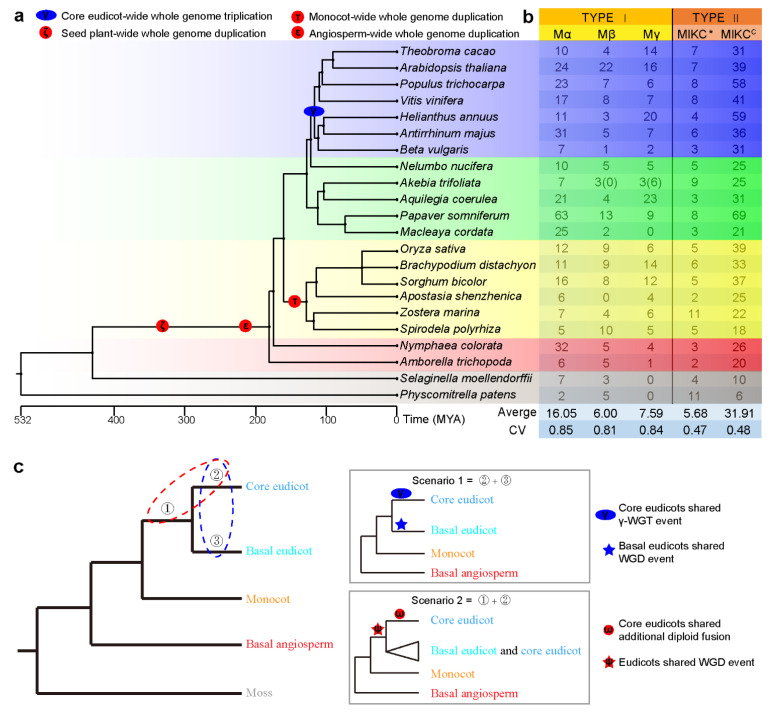
Diversification of MADS-box genes in land plants. (**a**) Phylogeny of 22 plant species. The topology of the tree was referenced from the TimeTree database, and four ancestral polyploidy events in seed plants were marked according to a previous report [44]. (**b**) Total number and classification of MADS-box genes in 22 plant species. CV: coefficient of variation. (**c**) Hypothetical duplicate genes of MADS-box origination from ancestral polyploidization events.

**Figure 5 genes-13-01777-f005:**
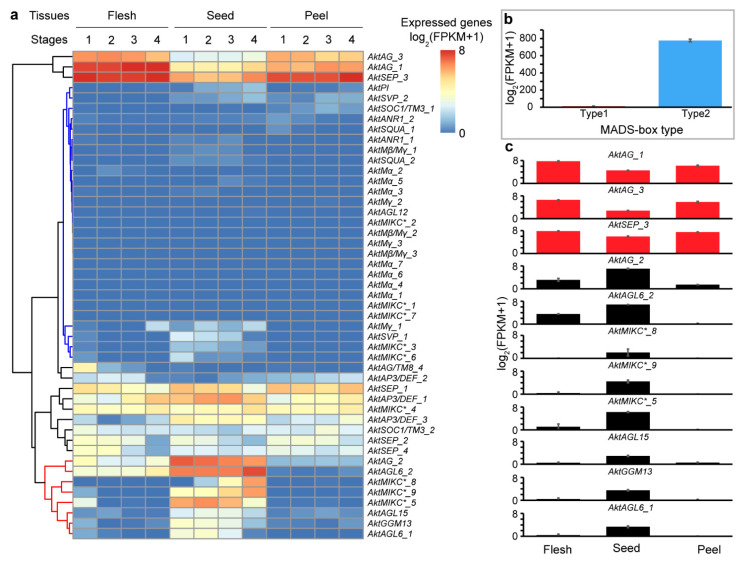
Expression levels of 47 MADS-box genes in different tissues and developmental stages. (**a**) Cluster results of gene expressions, FPKM represent fragments per kilobase of transcript per million fragments mapped. (**b**) Total expression levels of type I and type II genes, respectively. (**c**) Expressions of representative gene in three tissues.

**Table 1 genes-13-01777-t001:** Characteristics of the identified MADS-box gene families from the *A. trifoliata* genome.

Gene_Name	MADS-Box Type	Modified Name	Chromosome	Start Locus	Gene Length	Exon	AA	PI	MW (kDa)
*EVM0003126*	TypeⅠ:Mα	*AktMα_1*	chr2	9340686	618	2	192	8.23	21.38
*EVM0007393*	TypeⅠ:Mα	*AktMα_2*	chr4	13263148	627	2	197	8.65	22.4
*EVM0008161*	TypeⅠ:Mα	*AktMα_3*	chr3	40316320	705	1	234	10	26.3
*EVM0008221*	TypeⅠ:Mα	*AktMα_4*	chr4	13240370	1167	1	388	4.67	42.55
*EVM0009312*	TypeⅠ:Mα	*AktMα_5*	chr8	6769455	4439	2	204	9.2	23.55
*EVM0019163*	TypeⅠ:Mα	*AktMα_6*	chr4	13227911	6676	2	182	7.58	20.71
*EVM0019453*	TypeⅠ:Mα	*AktMα_7*	chr8	45540661	420	1	139	9.6	15.98
*EVM0009117*	TypeⅠ:Mβ/Mγ	*AktMβ_1*	chr13	27782386	3336	2	345	4.84	38.81
*EVM0013722*	TypeⅠ:Mβ/Mγ	*AktMβ_2*	chr16	31917540	981	1	326	5.3	37.12
*EVM0016918*	TypeⅠ:Mβ/Mγ	*AktMβ_3*	chr15	901697	951	1	316	8.29	36.85
*EVM0007317*	TypeⅠ:Mγ	*AktMγ_1*	chr8	15288657	782	1	251	6.8	28.49
*EVM0011701*	TypeⅠ:Mγ	*AktMγ_2*	chr4	4837008	663	1	220	5.52	25.59
*EVM0019125*	TypeⅠ:Mγ	*AktMγ_3*	chr16	32364845	3328	3	274	9.54	31.63
*EVM0002119*	TypeⅡ:MIKC*	*AktMIKC*_1*	chr15	11731497	2402	3	87	10.41	9.64
*EVM0008992*	TypeⅡ:MIKC*	*AktMIKC*_2*	Contig00271	36366	1726	2	82	10.81	9.1
*EVM0014464*	TypeⅡ:MIKC*	*AktMIKC*_3*	chr2	7636528	6189	13	420	4.7	47.72
*EVM0015933*	TypeⅡ:MIKC*	*AktMIKC*_4*	chr3	6814225	9410	12	352	7.41	40.16
*EVM0001820*	TypeⅡ:MIKC*	*AktMIKC*_5*	chr13	2907447	7905	11	326	7.54	36.95
*EVM0003153*	TypeⅡ:MIKC*	*AktMIKC*_6*	chr4	1021642	360	2	81	11.42	9.39
*EVM0007460*	TypeⅡ:MIKC*	*AktMIKC*_7*	chr4	14680429	6077	10	341	6.23	38.85
*EVM0011606*	TypeⅡ:MIKC*	*AktMIKC*_8*	chr2	12505863	4919	10	297	6.68	34.14
*EVM0012489*	TypeⅡ:MIKC*	*AktMIKC*_9*	chr3	42029038	16357	11	287	6.72	32.76
*EVM0008434*	TypeⅡ:GGM13	*AktGGM13*	chr1	21901520	17282	7	239	8.46	27.92
*EVM0009546*	TypeⅡ:AG	*AktAG_1*	chr11	49371746	8551	8	256	9.59	29.72
*EVM0023593*	TypeⅡ:AG	*AktAG_2*	chr3	33675447	37442	7	196	9.69	22.94
*EVM0016249*	TypeⅡ:AG	*AktAG_3*	chr11	5816813	12572	9	226	9.61	26.08
*EVM0004910*	TypeⅡ:AG/TM8	*AktAG/TM8*	chr9	26578608	5775	8	203	10.53	23.62
*EVM0008270*	TypeⅡ:AGL12	*AktAGL12*	chr5	25729606	7325	8	238	7.84	27.26
*EVM0012263*	TypeⅡ:AGL15	*AktAGL15*	chr14	89447	8175	8	248	8.29	28.48
*EVM0004219*	TypeⅡ:AGL6	*AktAGL6_1*	chr12	32312628	12346	8	243	9.16	27.9
*EVM0022986*	TypeⅡ:AGL6	*AktAGL6_2*	chr10	27960166	13564	8	243	9.21	27.78
*EVM0013426*	TypeⅡ:ANR1	*AktANR1_1*	chr3	39522622	25664	8	235	9.76	26.98
*EVM0020447*	TypeⅡ:ANR1	*AktANR1_2*	chr4	390010	25694	8	240	10.38	27.67
*EVM0022723*	TypeⅡ:AP3/DEF	*AktAP3/DEF_1*	chr2	3614814	2513	7	225	9.57	26.05
*EVM0011008*	TypeⅡ:AP3/DEF	*AktAP3/DEF_2*	chr4	42333387	6291	7	225	7.78	25.98
*EVM0016520*	TypeⅡ:AP3/DEF	*AktAP3/DEF_3*	chr2	3598851	2268	7	227	10.42	26.52
*EVM0017990*	TypeⅡ:PI	*AktPI*	chr3	36571931	4002	7	212	9.48	24.81
*EVM0006618*	TypeⅡ:SEP	*AktSEP_1*	chr9	21528809	25656	8	246	8.45	28.3
*EVM0009403*	TypeⅡ:SEP	*AktSEP_2*	chr5	8341522	80167	5	188	7.34	21.22
*EVM0012649*	TypeⅡ:SEP	*AktSEP_3*	chr7	24138805	37115	9	254	8.93	29.22
*EVM0001967*	TypeⅡ:SEP	*AktSEP_4*	chr7	33179065	15121	8	246	7.24	27.96
*EVM0001926*	TypeⅡ:SOC1/TM3	*AktSOC1/TM3_1*	chr12	32400886	20703	8	231	7.94	26.59
*EVM0003932*	TypeⅡ:SOC1/TM3	*AktSOC1/TM3_2*	chr12	32427991	30375	19	545	9.19	62.53
*EVM0013498*	TypeⅡ:SQUA	*AktSQUA_1*	chr7	33206678	19735	8	260	9.83	30.07
*EVM0021295*	TypeⅡ:SQUA	*AktSQUA_2*	chr3	10011542	36166	4	126	10.92	14.97
*EVM0004256*	TypeⅡ:SVP	*AktSVP_1*	chr4	10882672	11337	8	251	6.54	28.63
*EVM0015068*	TypeⅡ:SVP	*AktSVP_2*	chr2	6237389	9183	9	225	6.96	25.42

Abbreviations: AA, amino acid; PI, isoelectric point; MW, molecular weight.

## Data Availability

All data analyzed during this study are included in the manuscript and Appendix A, and transcriptomic data of *A. trifoliata* fruit tissues have been deposited in the National Center for Biotechnology Information (NCBI) database (https://www.ncbi.nlm.nih.gov/sra under the accession numbers SAMN16551931–33, SAMN16551934–36, SAMN16551937–39 and SAMN16551940–42, accessed on 28 June 2022).

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
