# Peer review of "Characterization of the MADS-Box Gene Family in Akebia trifoliata and Their Evolutionary Events in Angiosperms"

_genes, 2022, doi:10.3390/genes13101777_

Round 1

Reviewer 1 Report

Authors have presented the work on characterizing the MADS BOX gene family in Akebia trifoliata which is an important member of the basal eudicot group.

Following are few points to consider:

·         Give a context of the work done in the beginning of Abstract before the results are summarised.

·         Start with the introduction of the system studied followed by the MADS BOX TF

Overall article needs minor corrections and appears to add value to the readers and specifically to researchers in this field.

Author Response

Response to reviewer 1: Thank you for your positive and valuable comments. We have revised and added the background of study in Abstract according to your suggestion.

Reviewer 2 Report

Zhong et al characterise the MADS-Box gene family in Akebia trifoliata, a basal eudicot. By conducting sequence similarity searches, the authors detect 13 type I and 34 type II MADS-box genes. In a descriptive manner, the authors highlight their chromosome location, protein domain structure and inter-species synteny. Further, the authors investigate the abundance of MADS-box sub-types and two possible evolutionary scenarios that might have led to the current number of MADS-box genes in Akebia trifoliata. The authors close their manuscript by conducting an expression analysis in three fruit related tissues and four different stages to highlight expression but no further expression correlation.

Major concern:

There are major concerns regarding the method section. Important information is missing so that the findings are not reproducable. E.g. the complete part of how the genome of A. trifoliata was re-assembled and how genes were annotated on this assembly is missing. There are sparse information on the Bioproject page about the assembly software (https://www.ncbi.nlm.nih.gov/assembly/GCA_017979445.1; Canu v. 1.5; MHAP v. 2.1; Wtdbg v. 2.0; MUMmer v. 4.0.0) and gene annotations are not mentioned nor given.

Software versions are not mentioned in the manuscript. It is not mentioned how the different duplication types were exactly called by the synteny analysis. Again, version numbers of HMMER, MCScanX and TBTools are missing. The authors do not provide the necessary details of RNAseq analysis steps to reproduce their findings, e.g. HiSat2 mapping options, how FPKM were counted from BAM files and which DESeq2 settings were applied on the count matrix.

Further it is not clear if pre-annotated MADS-box genes or the same search pipeline as for A. trifoliata was used to fetch MADS-box genes for 22 plant species, to reconstruct MADS-box phylogeny.

Why the authors have used an e-value of 1e-5 to detect Pfam domains for MADS-box gene identification and not e.g. the gathering cutoff (pfam.xfam.org/help?tab=helpScoresBlock)?

The authors state on lines 216-219 "Compared with A. trifoliata MADS-box genes from a previous report and the GenBank database, only a few genes (10) related to floral organ development were reported (Table S2), and most of the genes (37) identified in our study have never been detected before."

However, looking at a previous study (https://www.nature.com/articles/s41586-019-1693-2) it seems that A. trifoliata was among the analysed species, including whole-genome duplication prediction. I wonder, why the authors do not compare their findings which are now based on whole-genome data against the published transcriptome data.

Figure 2a: It is not clear, if this tree is a sub-tree from Figure 1 or how this tree was constructed.

Why a sliding window of 250kbp was used to define genomic blocks of MADS-box containing genes? How does this compare to other Ranunculales for which annotated genomes exist?

Author Response

Response to Major concern 1: Sorry for the missing information. The details of both assembling and annotating processes of A. trifoliata genome would be shown in our other manuscript, which is being under review in the other journal. In briefly, the genome was assembled into into 1,553 contigs (N50 = 1.57 Mb, scaffolds N50 = 35.75 Mb and LTR Assembly Index score = 11.90) and the combined strategy including homology, RNA-seq and de novo-based methods were used to predict the Protein-coding genes of A. trifoliata. In addition, the gene annotations have been deposited in the Genome Warehouse of the National Genomics Data Center under the accession number GWHBISH00000000 (https://ngdc.cncb.ac.cn/search/?dbId=gwh&q=GWHBISH00000000&page=1). Finally, we have also added annotations information to the Materials and Methods section of the revision. Please check it again.

Response to Major concern 2: Sorry for the missing information. All software versions have been added to the Materials and Methods in the revision according to your comments. In addition, the details of synteny and RNA-seq analysis have also been added in the revision.

Response to Major concern 3: Yes, the same search method was used to fetch MADS-box genes for 22 plant species. We have added the related description in the methods.

Response to Major concern 4: The e-value of 1e-5 in hmmsearch is widely used in many researches (Lavrinienko et al., 2020. doi:10.1038/s41597-020-00656-2; Qiao et al. 2018. doi: 10.3389/fpls.2018.01395). In addition, we have compared and checked the results with or without e-value cutoff in A. trifoliata. The lowest output values of matched gene from hmmsearch result in A. trifolita is e-value of 1.30e-11 and bit core of 44.2, which is far from 1e-5 e-value threshold or 25 bit score GA. So whether to use e-value or bit score GA threshold does not affect the results of MAD-box gene identification in A. trifolita.

Response to Major concern 5: Thank you for providing valuable infomation. We previously focused on the MADS-box family studies in A. trifolita rather than the WGD, so we did not sufficiently concerned this study. In OneKP study, an common WGD event, called PASOβthat is phylogenetic close to the core eudicot common WGD(t) called ARTHγ(Supplementary Table 2 in OneKP), was detected in A. trifolita and many other basal eudicots. The authors of OneKP also cited another study (Aköz and Magnus Nordborg, 2019. doi:10.1186/s13059-019-1888-8) in Prior Publication about PASOβ (Supplementary Table 2 in OneKP), which is supported our two-step WGD process found in MADS-box genes. About MADS-box family in OneKP study, only a general statistics of MADS-box genes in Extended Data Fig. 1,2. It's hard for us to directly compare this part of results in A. trifolita. Gene-family nucleotide and amino acid FASTA files of OneKP study were provided in http://jlmwiki.plantbio.uga.edu/onekp/v2/, but we are unable to visit this website after multiple attempts.

In the revision, we have added some discussions about MADS-box family and WGD event results of OneKP study in the Discussion section.

Response to Major concern 6: Sorry for the unclear description. The tree in Figure 2a was independently constructed only by A. trifolita MADS-box genes with the same method of the tree in Figure 1. We have added the related information in the Materials and methods.

Response to Major concern 7: In this study, the 250kbp sliding window was only applied to observe gene cluster of MADS-box family. The definition of sliding window is based on the previous report of model plant Medicago truncatula (Ameline-Torregrosa et al. 2008. doi: 10.1104/pp.107.104588). The number of sequenced Ranunculales genomes is not much and this definition is only used in the gene cluster analysis of specific gene family. To our knowledge, the similar studies and application scenarios are rare in other Ranunculales.